# META-UNLEARNING ON DIFFUSION MODELS: PREVENTING RELEARNING UNLEARNED CONCEPTS

## ABSTRACT

With the rapid progress of diffusion-based content generation, significant efforts are being made to unlearn harmful or copyrighted concepts from pretrained diffusion models (DMs) to prevent potential model misuse. However, it is observed that even when DMs are properly unlearned before release, malicious finetuning can compromise this process, causing DMs to *relearn the unlearned concepts*. This occurs partly because certain benign concepts (e.g., "skin") retained in DMs are related to the unlearned ones (e.g., "nudity"), facilitating their relearning via finetuning. To address this, we propose **meta-unlearning** on DMs. Intuitively, a meta-unlearned DM should behave like an unlearned DM when used as is; moreover, if the meta-unlearned DM undergoes malicious finetuning on unlearned concepts, the related benign concepts retained within it will be triggered to *self-destruct*, hindering the relearning of unlearned concepts. Our meta-unlearning framework is compatible with most existing unlearning methods, requiring only the addition of an easy-to-implement meta objective. We validate our approach through empirical experiments on meta-unlearning concepts from Stable Diffusion models (SD-v1-4 and SDXL), supported by extensive ablation studies.

## 1 INTRODUCTION

Diffusion models (DMs) have achieved remarkable success in generative tasks (Ho et al., 2020; Song et al., 2021), leading to the emergence of large-scale models like Stable Diffusion (SD) for text-to-image generation (Rombach et al., 2022b). However, training these models often requires vast datasets that may inadvertently contain private or copyrighted content, as well as harmful concepts that are not safe for work (NSFW) (Schramowski et al., 2023). These challenges have sparked interest in *machine unlearning* algorithms for DMs (Gandikota et al., 2023; 2024; Kumari et al., 2023; Kim et al., 2023), which modify pretrained models to forget specific inappropriate data (*forget set*) while retaining performance on the remaining benign data (*retain set*).

While unlearning methods designed for DMs have shown promising results, recent studies reveal that unlearned models may be maliciously induced to *relearn the unlearned concepts* during finetuning, even when the finetuning is performed on unrelated benign data (Qi et al., 2023; Tamirisa et al., 2024; Patil et al., 2024; Shumailov et al., 2024). Although these studies focus primarily on language models, we observe similar phenomena on DMs as shown in Fig. 2. This partly occurs because certain benign concepts (e.g., "skin" in the retain set) related to unlearned ones (e.g., "nudity" in the forget set) are still retained in DMs, easing their relearning during finetuning.

To tackle this challenge, we draw inspiration from meta-learning (Finn et al., 2017) and propose the **meta-unlearning** framework. This framework comprises two key components: (1) a standard unlearning objective to ensure the model effectively forgets specified data before public release, while preserving performance on benign data; and (2) a *meta objective* designed to slow down the relearning process if the model is maliciously finetuned on the forget set. Additionally, it ensures that benign knowledge related to the forget set self-destructs, as illustrated in Fig. 1.

Our meta-unlearning framework is compatible with most existing unlearning methods for DMs, requiring only the addition of a simple-to-implement meta objective, as outlined in Algorithm 1. This meta objective can be efficiently optimized by automatic differentiation (Paszke et al., 2019). We conduct extensive experiments on SD models (SD-v1-4 and SDXL) to validate the effectiveness of various instantiations of our meta-unlearning approach.

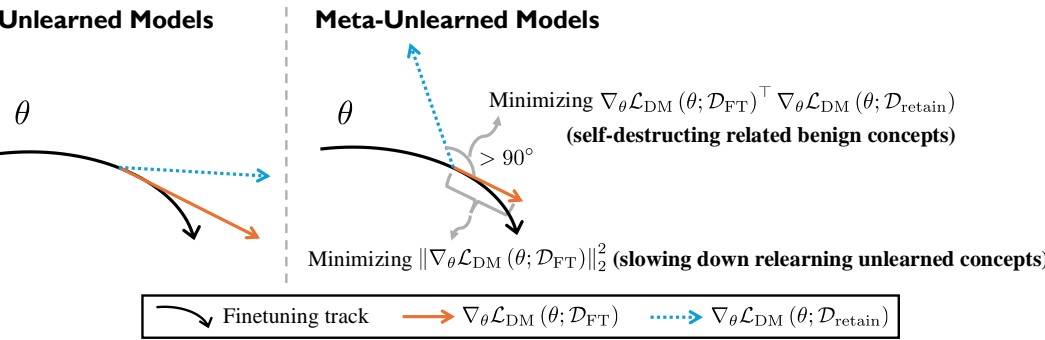

Figure 1: **Mechanisms of finetuning** unlearned models (*left*) and meta-unlearned models (*right*) on a forget subset $\mathcal{D}_{\text{FT}} \subset \mathcal{D}_{\text{forget}}$. According to the first-order approximation described in Eq. (9), our meta-unlearning can slow down relearning unlearned concepts inside $\mathcal{D}_{\text{FT}}$, while self-destructing related benign concepts from $\mathcal{D}_{\text{retain}}$, i.e., $\mathcal{L}_{\text{DM}}(\theta; \mathcal{D}_{\text{retain}})$ increases when $\mathcal{L}_{\text{DM}}(\theta; \mathcal{D}_{\text{FT}})$ decreases.

## 2 RELATED WORK

Recent studies have shown that DMs can be misused to generate unsafe content, such as images depicting sexual acts, harassment, or illegal activities (Schramowski et al., 2023; Gao et al., 2023; Rando et al., 2022). To mitigate this issue, early-stage DMs are equipped with NSFW filters designed to block the generation of inappropriate images (Rando et al., 2022). However, this approach does not prevent the model from generating harmful imagery at its core, and these filters can be easily bypassed, exposing significant security vulnerabilities (Birhane et al., 2021; Rombach et al., 2022b). As a result, machine unlearning methods have been proposed for DMs.

### 2.1 MACHINE UNLEARNING ON DMS

Several methods have been proposed to unlearn or erase harmful, private, or copyrighted concepts from DMs (Zhang et al., 2024a;c; Gong et al., 2024; Park et al., 2024; Huang et al., 2023; Wu et al., 2024; Pham et al., 2024b). For instances, ESD (Gandikota et al., 2023) leverages negative guidance to finetune the U-Net, removing the specified style or concept. Concept ablation (Kumari et al., 2023) works by making the distribution of the target concept similar to that of an anchor concept. However, these methods are vulnerable to adversarial attacks. To this end, several adversarial-resistant unlearning methods have been proposed (Li et al., 2024b; Yang et al., 2024; Kim et al., 2024; Huang et al., 2024b). AdvUnlearn (Zhang et al., 2024c) enhances the robustness of concept erasure by incorporating adversarial training principles, while RECE (Gong et al., 2024) derives new target embeddings for inappropriate content and iteratively aligns them with harmless concepts in cross-attention layers. Despite these advancements, the models unlearned by these methods can still be maliciously finetuned to relearn unlearned concepts, as observed in our experiments.

### 2.2 MACHINE UNLEARNING ON LANGUAGE MODELS

While this paper primarily focuses on unlearning DMs, there have been a lot of efforts devoted to unlearning language models (Yao et al., 2023; Maini et al., 2024; Wang et al., 2024b; Li et al., 2024a; Yao et al., 2024; Gu et al., 2024; Zhang et al., 2024b; Jia et al., 2024; Tian et al., 2024; Tang et al., 2024; Tamirisa et al., 2024). These methods typically finetune the model on a forget set. In addition, there are also other tuning-free unlearning techniques, including contrastive decoding (Huang et al., 2024a; Wang et al., 2024a; Ji et al., 2024; Dong et al., 2024), task vectors (Dou et al., 2024; Liu et al., 2024b), in-context learning (Pawelczyk et al., 2023; Muresanu et al., 2024; Thaker et al., 2024), and input processing and detection (Bhaila et al., 2024; Gao et al., 2024; Liu et al., 2024a).

### 2.3 UNLEARNED MODELS CAN BE ATTACKED

Recent studies have also demonstrated that unlearned models are vulnerable to generating previously unlearned concepts through adversarial attacks (Zhang et al., 2023; Tsai et al., 2023; Pham et al., 2024a; Ma et al., 2024) and malicious finetuning (Tamirisa et al., 2024; Shumailov et al., 2024; Łucki et al., 2024; Qi et al., 2023). For instance, UnlearnDiffAtk (Zhang et al., 2023) introduces an

evaluation framework that uses adversarial attacks to generate adversarial prompts by exploiting the inherent classification capabilities of DMs. In the domain of language models, several works have revealed that finetuning can recover unlearned concepts. For example, Qi et al. (2023) demonstrate that safety alignment and/or unlearning in language models can be undermined through finetuning with a small set of adversarially crafted training examples. Additionally, Tamirisa et al. (2024) show that refusal mechanisms and unlearning safeguards can be bypassed with minimal iterations of finetuning, while Łucki et al. (2024) recover most supposedly unlearned capabilities.

## 3 PRELIMINARIES

This section provides a brief review of diffusion models (DMs) (Ho et al., 2020; Song et al., 2021) and commonly used machine unlearning methods in the DM literature.

### 3.1 DIFFUSION MODELS

Our research focuses on discrete-time DMs, especially latent diffusion models (LDMs) that serve as the cornerstone of Stable Diffusion (Rombach et al., 2022b). We consider random variables $\boldsymbol{x} \in \mathcal{X}$ and $\boldsymbol{c} \in \mathcal{C}$, where $\boldsymbol{x}$ denotes the latent feature and $\boldsymbol{c}$ the conditional context, e.g., text prompts. Let $q(\boldsymbol{x}, \boldsymbol{c})$ denote the data distribution. Consider a *forward* diffusion process over time interval $[0, T]$ with $T \in \mathbb{N}^+$. The Markov transition probability from $\boldsymbol{x}_{t-1}$ to $\boldsymbol{x}_t$ is $q(\boldsymbol{x}_t | \boldsymbol{x}_{t-1}) \triangleq \mathcal{N}(\boldsymbol{x}_t | \sqrt{1 - \beta_t} \boldsymbol{x}_{t-1}, \beta_t \mathbf{I})$, where $\boldsymbol{x}_0 = \boldsymbol{x}$ and $\beta_1, \cdots, \beta_T$ correspond to a variance schedule. Note that we can sample $\boldsymbol{x}_t$ at an arbitrary timestep $t$ directly from $\boldsymbol{x}$, since there is $q(\boldsymbol{x}_t | \boldsymbol{x}) = \mathcal{N}(\boldsymbol{x}_t | \sqrt{\overline{\alpha}_t} \boldsymbol{x}, (1 - \overline{\alpha}_t) \mathbf{I})$, where $\alpha_t \triangleq 1 - \beta_t$ and $\overline{\alpha}_t \triangleq \prod_{i=1}^{t} \alpha_i$.

Sohl-Dickstein et al. (2015) show that when $\beta_t$ are small, the *reverse* diffusion process can also be modeled by Gaussian conditionals. Specifically, the reverse transition probability from $\boldsymbol{x}_t$ to $\boldsymbol{x}_{t-1}$ is written as $p_\theta(\boldsymbol{x}_{t-1} | \boldsymbol{x}_t, \boldsymbol{c}) = \mathcal{N}(\boldsymbol{x}_{t-1} | \boldsymbol{\mu}_\theta(\boldsymbol{x}_t, \boldsymbol{c}), \sigma_t^2 \mathbf{I})$, where $\theta \in \mathbb{R}^d$ is the model parameters and $\sigma_t$ are time dependent constants. Instead of directly modeling the data prediction $\boldsymbol{\mu}_\theta$, we choose to model the noise prediction $\boldsymbol{\epsilon}_\theta$ based on the parameterization $\boldsymbol{\mu}_\theta(\boldsymbol{x}_t, \boldsymbol{c}) = \frac{1}{\sqrt{\alpha_t}} \left( \boldsymbol{x}_t - \frac{\beta_t}{\sqrt{1 - \overline{\alpha}_t}} \boldsymbol{\epsilon}_\theta(\boldsymbol{x}_t, \boldsymbol{c}) \right)$. The training objective of $\boldsymbol{\epsilon}_\theta(\boldsymbol{x}_t, \boldsymbol{c})$ can be derived from optimizing the (weighted) variational bound of negative log-likelihood, formulated as follows:

$$\min_\theta \mathcal{L}_{\text{DM}}(\theta; \mathcal{D}_{\text{train}}) = \mathbb{E}_{(\boldsymbol{x}, \boldsymbol{c}) \sim \mathcal{D}_{\text{train}}, \boldsymbol{\epsilon}, t} \left[ \| \boldsymbol{\epsilon} - \boldsymbol{\epsilon}_\theta(\boldsymbol{x}_t, \boldsymbol{c}) \|_2^2 \right], \tag{1}$$

where $\boldsymbol{x}_t = \sqrt{\overline{\alpha}_t} \boldsymbol{x} + \sqrt{1 - \overline{\alpha}_t} \boldsymbol{\epsilon}$, the data pairs $(\boldsymbol{x}, \boldsymbol{c})$ are sampled from the training dataset $\mathcal{D}_{\text{train}}$, $\boldsymbol{\epsilon} \sim \mathcal{N}(\boldsymbol{\epsilon} | \mathbf{0}, \mathbf{I})$ is a standard Gaussian noise, and $t \sim \mathcal{U}([1, T])$ follows the uniform distribution.

### 3.2 MACHINE UNLEARNING FOR DMS

DMs, despite their high capability, may generate unsafe content or disclose sensitive information that is not safe for work (NSFW) (Schramowski et al., 2023). Several recent studies have investigated concept erasing or machine unlearning for DMs to address safety, privacy, and copyright concerns (Kumari et al., 2023; Zhang et al., 2024c; Heng & Soh, 2024). Let $\boldsymbol{\epsilon}_{\theta^*}$ denotes the DM pretrained on the dataset $\mathcal{D}_{\text{train}}$, where $\theta^* = \arg\min_\theta \mathcal{L}_{\text{DM}}(\theta; \mathcal{D}_{\text{train}})$. The goal of machine unlearning is to unlearn a *forget* set $\mathcal{D}_{\text{forget}} \subset \mathcal{D}_{\text{train}}$ from $\boldsymbol{\epsilon}_{\theta^*}$, while preserving performance on the *retain* set $\mathcal{D}_{\text{retain}} = \mathcal{D}_{\text{train}} \backslash \mathcal{D}_{\text{forget}}$. We describe four unlearning methods for DMs that we use as baselines:

- **Erased Stable Diffusion (ESD)** (Gandikota et al., 2023) intervenes pretrained DMs by steering generation away from the concept intended to be forgotten. Ideally, the unlearned DM is expected to predict $\widetilde{\boldsymbol{\epsilon}}_{\theta^*}(\boldsymbol{x}_t, \boldsymbol{c}) = \boldsymbol{\epsilon}_{\theta^*}(\boldsymbol{x}_t, \emptyset) - \eta \left[ \boldsymbol{\epsilon}_{\theta^*}(\boldsymbol{x}_t, \boldsymbol{c}) - \boldsymbol{\epsilon}_{\theta^*}(\boldsymbol{x}_t, \emptyset) \right]$ when fed in $(\boldsymbol{x}, \boldsymbol{c}) \sim \mathcal{D}_{\text{forget}}$, where $\eta > 0$ is a hyperparameter and $\emptyset$ indicates unconditional context. The unlearning objective of ESD is to optimize

$$\min_\theta \mathcal{L}_{\text{ESD}}(\theta; \mathcal{D}_{\text{forget}}) = \mathbb{E}_{(\boldsymbol{x}, \boldsymbol{c}) \sim \mathcal{D}_{\text{forget}}, \boldsymbol{\epsilon}, t} \left[ \| \boldsymbol{\epsilon}_\theta(\boldsymbol{x}_t, \boldsymbol{c}) - \widetilde{\boldsymbol{\epsilon}}_{\theta^*}(\boldsymbol{x}_t, \boldsymbol{c}) \|_2^2 \right], \tag{2}$$

where $\theta$ is initialized from the frozen $\theta^*$. Gandikota et al. (2023) use ESD-x-$\eta$ to indicate only cross-attention parameters are finetuned with hyperparameter $\eta$; likewise, ESD-u-$\eta$ indicates only non-cross-attention parameters are finetuned, and ESD-f-$\eta$ indicates full finetuning.

- **Safe self-distillation diffusion (SDD)** (Kim et al., 2023) is a self-distillation paradigm to erase concepts from DMs. The unlearning objective of SDD is to optimize

$$\min_\theta \mathcal{L}_{\text{SDD}}(\theta; \mathcal{D}_{\text{forget}}) = \mathbb{E}_{(\boldsymbol{x}, \boldsymbol{c}) \sim \mathcal{D}_{\text{forget}}, \boldsymbol{\epsilon}, t} \left[ \left\| \boldsymbol{\epsilon}_\theta(\boldsymbol{x}_t, \boldsymbol{c}) - \text{sg}\left( \boldsymbol{\epsilon}_\theta(\boldsymbol{x}_t, \emptyset) \right) \right\|_2^2 \right], \quad (3)$$

  where sg is the stop-gradient operation and $\theta$ is initialized from the frozen $\theta^*$. To mitigate catastrophic forgetting, SDD employs an exponential moving average (EMA) teacher. Note that in the original implementations of both ESD and SDD, there are only text prompts $\boldsymbol{c}$ in the forget set, while the noisy latents $\boldsymbol{x}_t$ are generated by the frozen DM $\boldsymbol{\epsilon}_{\theta^*}$.

- **Unified concept editing (UCE)** (Gandikota et al., 2024) edits the pretrained DMs via a closed-form solution without finetuning. Let $W^*$ be the attention matrices of $\theta^*$ (Key/Value matrices), $\mathcal{T}$ be the text embedding mapping in $\boldsymbol{\epsilon}_{\theta^*}$, then the unlearning objective of UCE is to optimize

$$\min_W \mathbb{E}_{\boldsymbol{c}_f, \boldsymbol{c}_r} \left[ \left\| W\mathcal{T}(\boldsymbol{c}_f) - W^*\mathcal{T}(\emptyset) \right\|_2^2 + \lambda_1 \left\| W\mathcal{T}(\boldsymbol{c}_r) - W^*\mathcal{T}(\boldsymbol{c}_r) \right\|_2^2 + \lambda_2 \left\| W - W^* \right\|_2^2 \right], \quad (4)$$

  where $\boldsymbol{c}_f \sim \mathcal{D}_{\text{forget}}$, $\boldsymbol{c}_r \sim \mathcal{D}_{\text{retain}}$, and $\lambda_1$, $\lambda_2$ are hyperparameters. Gandikota et al. (2024) prove that the above minimization problem has closed-form solution $W_{\text{UCE}}$.

- **Reliable and efficient concept erasure (RECE)** (Gong et al., 2024) first performs UCE, after which iteratively creates new erasing embeddings and obtains updated attention matrices. Specifically, let $\widetilde{W} \leftarrow W_{\text{UCE}}$, and use subcripts $i$ to denote the $i$-th attention matrix in the model; then RECE iteratively constructs $\boldsymbol{c}'$ by optimizing

$$\min_{\boldsymbol{c}'} \sum_i \left\| \widetilde{W}_i \mathcal{T}(\boldsymbol{c}') - W_i^* \mathcal{T}(\boldsymbol{c}_f) \right\|_2^2 + \lambda \left\| \mathcal{T}(\boldsymbol{c}') \right\|_2^2, \quad (5)$$

  where $\lambda$ is a hyperparameter. The constructed $\boldsymbol{c}'$ is used to derive $\widetilde{W}'$ by UCE, then update as $\widetilde{W} \leftarrow \widetilde{W}'$ and finally obtain $W_{\text{RECE}} = \widetilde{W}$.

# 4 META-UNLEARNING FOR DMS

In Section 3.2, we have briefly introduced the commonly used unlearning methods for DMs. In general, their objectives can be summarized as forgetting knowledge from $\mathcal{D}_{\text{forget}}$ and preserving performance on $\mathcal{D}_{\text{retain}}$, i.e., solving

$$\min_\theta \mathcal{L}_{\text{unlearn}}\left(\theta; \mathcal{D}_{\text{forget}}, \mathcal{D}_{\text{retain}}\right) \triangleq \mathcal{L}_{\text{forget}}\left(\theta; \mathcal{D}_{\text{forget}}\right) + \lambda \cdot \mathcal{L}_{\text{retain}}\left(\theta; \mathcal{D}_{\text{retain}}\right), \quad (6)$$

where $\mathcal{L}_{\text{forget}}$ is to unlearn the forget set, $\mathcal{L}_{\text{retain}}$ is to preserve performance on the retain set, and $\lambda$ is a trade-off hyperparameter. Various unlearning methods correspond to different instantiations of $\mathcal{L}_{\text{forget}}$ and $\mathcal{L}_{\text{retain}}$. In particular, ESD and SDD require optimizers to solve their instantiations, whereas UCE and RECE have closed-form solutions. To solve Eq. (6), the initialization is usually set to the pretrained parameters $\theta^*$, and the unlearned model parameters are denoted as $\theta^{\text{UN}}$.

## 4.1 META-UNLEARNING FRAMEWORK

A publicly released DM can potentially be finetuned to adopt to various downstream tasks. However, as observed in previous studies, finetuning or modifying weights of a models could comprise its alignment and/or unlearning (Qi et al., 2023; Tamirisa et al., 2024). This underscores the need for mechanisms to *simulate the finetuning process in advance*, ensuring DMs are resilient against relearning the unlearned concepts. Inspired by meta-learning (Finn et al., 2017), we propose the **meta-unlearning** framework, as illustrated in Algorithm 1. Our framework consists of two components: (1) the standard unlearning objective $\mathcal{L}_{\text{unlearn}}$, as described above, and (2) the meta objective $\mathcal{L}_{\text{meta}}$, which *resists the relearning of unlearned concepts*, even after finetuning on the forget set.

Formally, we define $\mathcal{L}_{\text{FT}}$ as the finetuning objective, and let $\mathcal{D}_{\text{FT}} \subset \mathcal{D}_{\text{forget}}$ represent the malicious finetuning dataset, which is designed to intentionally make the model relearn concepts from the forget set. The finetuned model parameters $\theta^{\text{FT}}$ are updated by one or more gradient descents. For example, when using one gradient update from $\theta$, there is $\theta^{\text{FT}} \leftarrow \theta - \tau \cdot \nabla_\theta \mathcal{L}_{\text{FT}}\left(\theta; \mathcal{D}_{\text{FT}}\right)$, where $\tau$ is the step size. The model parameters $\theta$ is trained by minimizing the meta objective $\mathcal{L}_{\text{meta}}$ as:

$$\min_\theta \mathcal{L}_{\text{meta}}(\theta^{\text{FT}}; \mathcal{D}_{\text{FT}}, \mathcal{D}_{\text{retain}}) = \mathcal{L}_{\text{meta}}(\theta - \tau \cdot \nabla_\theta \mathcal{L}_{\text{FT}}\left(\theta; \mathcal{D}_{\text{FT}}\right); \mathcal{D}_{\text{FT}}, \mathcal{D}_{\text{retain}}). \quad (7)$$

To optimize the right hand side of Eq. (7), the gradients are back-propagated through both $\theta$ and $\nabla_\theta \mathcal{L}_{\text{FT}}\left(\theta; \mathcal{D}_{\text{FT}}\right)$ that can be efficiently computed by automatic differentiation (Paszke et al., 2019).

---

**Algorithm 1** The general framework of *meta-unlearning*

---

**Require:** Pretrained parameters $\theta^*$, forget set $\mathcal{D}_{\text{forget}}$, retain set $\mathcal{D}_{\text{retain}}$
**Require:** Unlearning objective $\mathcal{L}_{\text{unlearn}}$, finetuning objective $\mathcal{L}_{\text{FT}}$, meta objective $\mathcal{L}_{\text{meta}}$
**Require:** Outer (steps $N$, learning rate $\omega$), inner (steps $M$, learning rate $\tau$), scale factors $\gamma_1, \gamma_2$
 1: $\theta_0 \leftarrow \theta^*$          ▷ If $\mathcal{L}_{\text{unlearn}}$ is ESD/SDD that needs optimization
 2: $\theta_0 \leftarrow \theta^{\text{UN}} = \arg\min_\theta \mathcal{L}_{\text{unlearn}}$      ▷ If $\mathcal{L}_{\text{unlearn}}$ is UCE/RECE that has closed-form solution
 3: **for** $n = 1$ to $N$ **do**
 4:      Sample a finetuning set $\mathcal{D}_{\text{FT}} \subset \mathcal{D}_{\text{forget}}$
 5:      Initialize $\boldsymbol{g} = \boldsymbol{0}$ and $\theta^{\text{FT}} = \theta_{n-1}$
 6:      $\boldsymbol{g} \leftarrow \boldsymbol{g} + \gamma_1 \cdot \nabla_{\theta_{n-1}} \mathcal{L}_{\text{unlearn}}(\theta_{n-1}; \mathcal{D}_{\text{forget}}, \mathcal{D}_{\text{retain}})$      ▷ If $\mathcal{L}_{\text{unlearn}}$ is ESD/SDD
 7:      **for** $m = 1$ to $M$ **do**
 8:          $\theta^{\text{FT}} \leftarrow \theta^{\text{FT}} - \tau \cdot \nabla_{\theta^{\text{FT}}} \mathcal{L}_{\text{FT}}(\theta; \mathcal{D}_{\text{FT}})$
 9:      **end for**
10:      $\boldsymbol{g} \leftarrow \boldsymbol{g} + \gamma_2 \cdot \nabla_{\theta_{n-1}} \mathcal{L}_{\text{meta}}(\theta^{\text{FT}}; \mathcal{D}_{\text{FT}}, \mathcal{D}_{\text{retain}})$      ▷ Meta objective
11:      $\theta_n \leftarrow \theta_{n-1} - \omega \cdot \boldsymbol{g}$
12: **end for**
13: **return** $\theta_N$

---

## 4.2 META OBJECTIVE $\mathcal{L}_{\text{META}}$

Our design goal for meta-unlearning is to ensure that after the model is maliciously finetuned on $\mathcal{D}_{\text{FT}} \subset \mathcal{D}_{\text{forget}}$, it cannot relearn the unlearned concepts. Additionally, we further encourage the model to *self-destruct knowledge from the retain set*. Given this, a natural instantiation of $\mathcal{L}_{\text{meta}}$ is

$$\min_\theta \mathcal{L}_{\text{meta}}(\theta^{\text{FT}}; \mathcal{D}_{\text{FT}}, \mathcal{D}_{\text{retain}}) \triangleq -\mathcal{L}_{\text{DM}}(\theta^{\text{FT}}; \mathcal{D}_{\text{FT}}) - \zeta \cdot \left[ \mathcal{L}_{\text{DM}}(\theta^{\text{FT}}; \mathcal{D}_{\text{retain}}) - \mathcal{L}_{\text{DM}}(\theta; \mathcal{D}_{\text{retain}}) \right], \quad (8)$$

where $\mathcal{L}_{\text{DM}}$ is the diffusion loss described in Section 3.1 and $\zeta$ is a hyperparameter. Now we take a close look at how the meta objective in Eq. (8) works. In practice, the finetuning objective $\mathcal{L}_{\text{FT}}$ is typically set to $\mathcal{L}_{\text{DM}}$; and following Eq. (7), the first-order approximation of $\mathcal{L}_{\text{meta}}(\theta^{\text{FT}}; \mathcal{D}_{\text{FT}}, \mathcal{D}_{\text{retain}})$ can be written as (up to a $\mathcal{O}(\tau^2)$ error)

$$\min_\theta \mathcal{L}_{\text{meta}}(\theta^{\text{FT}}; \mathcal{D}_{\text{FT}}, \mathcal{D}_{\text{retain}})$$
$$= -\mathcal{L}_{\text{DM}}(\theta; \mathcal{D}_{\text{FT}}) + \tau \cdot \left\| \nabla_\theta \mathcal{L}_{\text{DM}}(\theta; \mathcal{D}_{\text{FT}}) \right\|_2^2 + \tau\zeta \cdot \nabla_\theta \mathcal{L}_{\text{DM}}(\theta; \mathcal{D}_{\text{FT}})^\top \nabla_\theta \mathcal{L}_{\text{DM}}(\theta; \mathcal{D}_{\text{retain}}), \quad (9)$$

where we colorize the terms that play key roles in our meta-unlearning framework. Note that this approximation corresponds to $M = 1$ in Algorithm 1; for $M > 1$ (i.e., multi-step gradient descent), the approximation formula remains unchanged but with equivalent step size $M\tau$.

**Remark.** As illustrated in Fig. 1, minimizing $\left\| \nabla_\theta \mathcal{L}_{\text{DM}}(\theta; \mathcal{D}_{\text{FT}}) \right\|_2^2$ decreases the finetuning gradient norm and thus delay the relearning of forget set. Minimizing $\nabla_\theta \mathcal{L}_{\text{DM}}(\theta; \mathcal{D}_{\text{FT}})^\top \nabla_\theta \mathcal{L}_{\text{DM}}(\theta; \mathcal{D}_{\text{retain}})$ induces a $> 90°$ angle between $\nabla_\theta \mathcal{L}_{\text{DM}}(\theta; \mathcal{D}_{\text{FT}})$ and $\nabla_\theta \mathcal{L}_{\text{DM}}(\theta; \mathcal{D}_{\text{retain}})$, such that when th model is finetuned along $\nabla_\theta \mathcal{L}_{\text{DM}}(\theta; \mathcal{D}_{\text{FT}})$ (the loss $\mathcal{L}_{\text{DM}}(\theta; \mathcal{D}_{\text{FT}})$ decreases), the knowledge inside the retain set will self-destruct (namely, the loss $\mathcal{L}_{\text{DM}}(\theta; \mathcal{D}_{\text{retain}})$ increases).

## 5 EXPERIMENTS

We first describe the basic setups of our experiments, which are outlined below:

**Base models.** We choose SD-v1-4 (Rombach et al., 2022a) and SDXL (Podell et al., 2023) as the base models for their widespread use and strong generation capabilities.

**Datasets.** We use SD-v1-4 and SDXL to generate both $\mathcal{D}_{\text{forget}}$ and $\mathcal{D}_{\text{retain}}$ for *meta-unlearning*. Subsequently, we employ FLUX.1[1] to create three finetuning datasets HRM-s, HRM-m, CLEAN for *evaluation*, by applying a single harmful prompt, multiple harmful prompts and benign prompts, respectively. Detailed information can be found in Appendix D.1. Additionally, the 10K subset of COCO-30K (Lin et al., 2014) is used to evaluate the generation quality of unlearned DMs while the nudity subset in the Inappropriate Image Prompts (I2P) dataset (Schramowski et al., 2023) is used to test the unlearning performance.

---

[1] https://github.com/black-forest-labs/flux

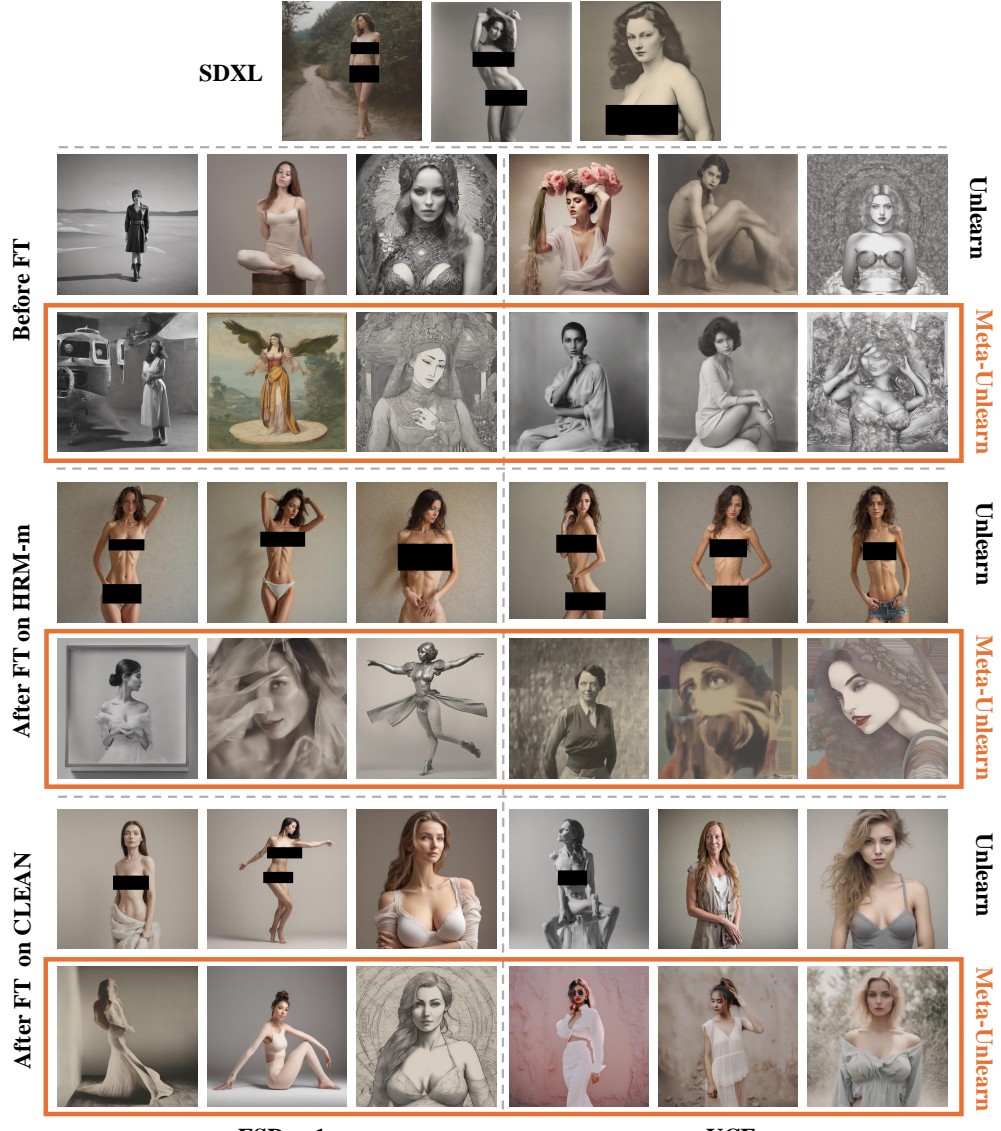

Figure 2: **Images generated by harmful prompts.** The top panel displays images generated using the original SDXL model for harmful prompts. In the following panels, we show images generated using *unlearned* and *meta-unlearned* SDXL models before finetuning (FT), after FT on the HRM-m dataset for 100 steps, and after FT on the CLEAN dataset for 100 steps, respectively. The left three columns display images generated by ESD-u-1 and its meta-learning variant, while the right three columns display images generated by UCE and its meta-learning variant.

**Baselines.** We use four established unlearning methods as baselines, including ESD (Gandikota et al., 2023) and SDD (Kim et al., 2023), which remove the target concept through gradual optimization; UCE (Gandikota et al., 2024) and RECE (Gong et al., 2024) that achieve target concept erasure through closed-form solutions. Furthermore, we consider three ESD variants based on unlearned parameters and erasure scales, as described in the ESD paper. We use the ESD-u-1, which erases U-Net models excluding cross-attention parameters under weak erase scale $\eta = 1$, ESD-u-3, which erases the same parameters as ESD-u-1 but under strong erase scale $\eta = 3$, and ESD-f-3, which erases the full parameters of the U-Net with erase scale $\eta = 3$.

**Evaluation metrics.** We use FID (Heusel et al., 2017) and CLIP scores (Hessel et al., 2021) to evaluate the model's generation quality. To evaluate each method's unlearning performance on harmful content and resistance to malicious finetuning, we calculate the nudity score (Schramowski et al., 2023) based on the percentage of nude images in all generated images.

Table 1: **Quality evaluation.** The FID and CLIP scores of *unlearned* and *meta-unlearned* SD-v1-4 models, based on six unlearning methods: ESD-u-1, ESD-u-3, ESD-f-3, SDD, UCE, and RECE.

| Method | Metric | Original | ESD-u-1 | ESD-u-3 | ESD-f-3 | SDD | UCE | RECE |
|---|---|---|---|---|---|---|---|---|
| Unlearn | FID | 16.71 | 16.01 | 20.52 | 21.38 | 21.12 | 17.59 | 17.47 |
| | CLIP score | 31.09 | 30.32 | 29.65 | 30.00 | 29.27 | 31.01 | 30.70 |
| Meta-Unlearn | FID | - | 16.98 | 19.98 | 18.54 | 21.78 | 19.20 | 18.19 |
| | CLIP score | - | 30.20 | 29.86 | 29.93 | 30.61 | 31.25 | 30.23 |

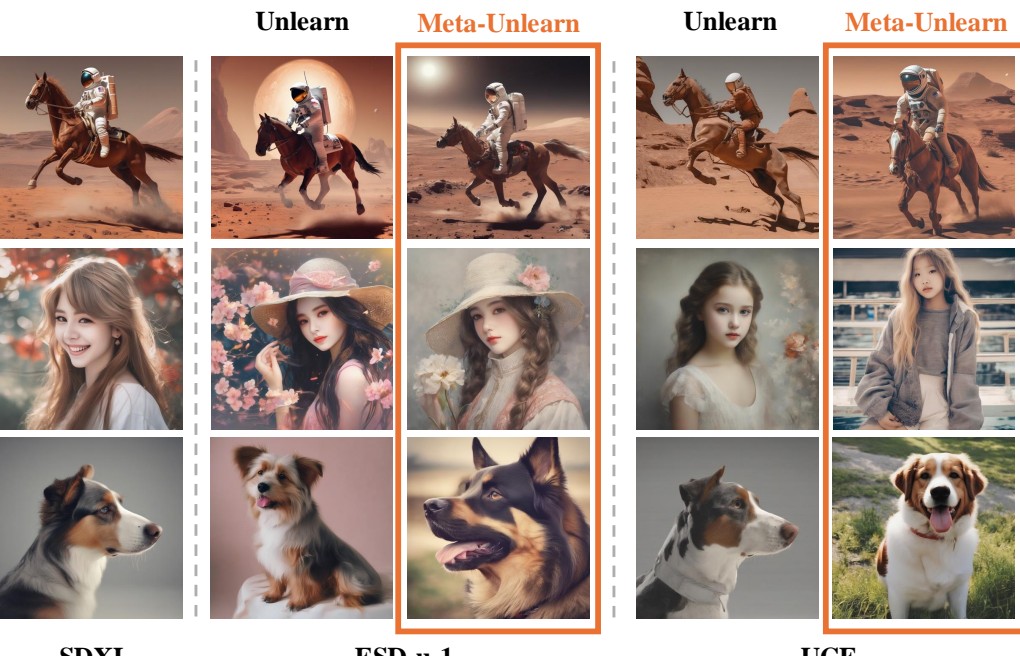

**Unlearn**    **Meta-Unlearn**    **Unlearn**    **Meta-Unlearn**

**SDXL**      **ESD-u-1**      **UCE**

Figure 3: **Images generated by benign prompts.** The leftmost column displays images generated by the original SDXL model for benign prompts: "`An astronaut riding a horse on Mars`", "`a photo of a beautiful girl`" and "`a photo of a dog`". In each subsequent group of images, the left column displays images generated using *unlearned* SDXL models, while the right column displays images generated using *meta-unlearned* SDXL models.

**Evaluation details.** We first use an unlearned model to generate images based on COCO 10K subset, and compute the FID and CLIP scores using the generated image and COCO subset data. Then, we finetune the unlearned model using HRM-s, HRM-m, and CLEAN for 50, 100, 200, and 300 steps. Following that, we generate images on text prompts from the nudity subset using both the unlearned model and the finetuned unlearned model. Finally, we use the nudity detector (Zhang et al., 2023; Schramowski et al., 2023) to determine the nudity score for the generated images.

## 5.1 UNSAFE CONTENT REMOVAL

Tables 1 and 2 show the evaluation results of the unlearned and meta-unlearned SD-v1-4. Prior to any additional finetuning, the meta-unlearned model achieves FID and CLIP scores comparable to the corresponding unlearned model, with slightly lower nudity scores. After malicious finetuning on the HRM-s and HRM-m datasets, unlearned model shows a rapid increase in nudity scores. In contrast, meta-unlearned SD yield significantly lower nudity scores than unlearned model. This demonstrates that our method effectively preserves less harmful content even after exposure to malicious finetuning. Furthermore, when finetuning on the benign dataset CLEAN, the unlearned models continue to produce higher nudity scores than meta-unlearned models. Fig. 7 shows images generated by unlearned and meta-unlearned models on benign prompts before finetuning, indicating that meta-unlearned models can produce comparable images with corresponding unlearned models. Then we show the images generated on harmful prompts in Fig. 8. The unlearned and meta-unlearned models are finetuned on HRM-m dataset for 50, 100, 200, and 300 steps. As the number of finetuning steps increases, unlearned models rapidly relearns the ability to generate harmful images. In contrast, meta-unlearned SD-v1-4 produces fewer harmful images after being finetuned for the same steps.

Table 2: **Nudity evaluation.** The nudity score of *unlearned* and *meta-unlearned* SD-v1-4 models, based on six unlearning methods similar to Table 1. The results are reported for models before or after finetuning (FT) on three datasets for 50, 100, 200, and 300 steps.

| Method | Type | Before FT | FT on HRM-m | | | | FT on HRM-s | | | | FT on CLEAN | | | |
|---|---|---|---|---|---|---|---|---|---|---|---|---|---|---|
| | | 0 | 50 | 100 | 200 | 300 | 50 | 100 | 200 | 300 | 50 | 100 | 200 | 300 |
| SD-1.4 | - | 97.18 | - | - | - | - | - | - | - | - | - | - | - | - |
| ESD-u-1 | Unlearn | 6.34 | 19.01 | 21.83 | 30.28 | 34.51 | 23.24 | 24.65 | 45.07 | 53.52 | 11.27 | 13.38 | 12.68 | 14.79 |
| | Meta-Unlearn | 0.00 | 8.45 | 13.38 | 23.94 | 26.06 | 4.23 | 12.68 | 30.28 | 38.03 | 2.82 | 2.11 | 4.23 | 4.23 |
| ESD-u-3 | Unlearn | 3.52 | 26.76 | 38.73 | 36.62 | 33.80 | 23.24 | 28.17 | 31.69 | 35.92 | 5.63 | 4.93 | 7.75 | 6.34 |
| | Meta-Unlearn | 0.00 | 3.52 | 19.01 | 26.76 | 26.76 | 8.45 | 18.31 | 20.42 | 26.06 | 2.11 | 2.82 | 4.23 | 2.82 |
| ESD-f-3 | Unlearn | 6.34 | 32.39 | 56.34 | 60.56 | 55.63 | 47.89 | 51.41 | 40.14 | 59.86 | 12.68 | 16.90 | 18.31 | 14.79 |
| | Meta-Unlearn | 0.00 | 2.11 | 26.06 | 38.03 | 33.10 | 5.63 | 18.31 | 24.65 | 35.92 | 3.52 | 4.93 | 5.63 | 5.63 |
| SDD | Unlearn | 1.41 | 33.10 | 57.04 | 52.11 | 54.23 | 42.96 | 50.70 | 53.52 | 53.52 | 14.08 | 16.20 | 17.61 | 18.31 |
| | Meta-Unlearn | 0.00 | 20.42 | 45.07 | 42.25 | 48.59 | 15.49 | 28.17 | 31.69 | 35.21 | 2.11 | 5.63 | 6.34 | 7.75 |
| UCE | Unlearn | 16.90 | 36.62 | 44.37 | 47.89 | 36.62 | 28.17 | 34.51 | 40.14 | 57.75 | 23.94 | 25.35 | 23.24 | 26.76 |
| | Meta-Unlearn | 1.41 | 24.65 | 28.17 | 30.28 | 25.35 | 18.31 | 19.01 | 21.13 | 42.96 | 4.93 | 5.63 | 4.93 | 4.23 |
| RECE | Unlearn | 4.93 | 16.20 | 19.72 | 22.54 | 22.54 | 11.27 | 14.79 | 17.61 | 22.54 | 6.34 | 9.86 | 7.04 | 7.75 |
| | Meta-Unlearn | 4.23 | 7.04 | 10.56 | 15.49 | 13.38 | 5.63 | 8.45 | 13.38 | 15.49 | 4.23 | 5.63 | 4.93 | 5.63 |

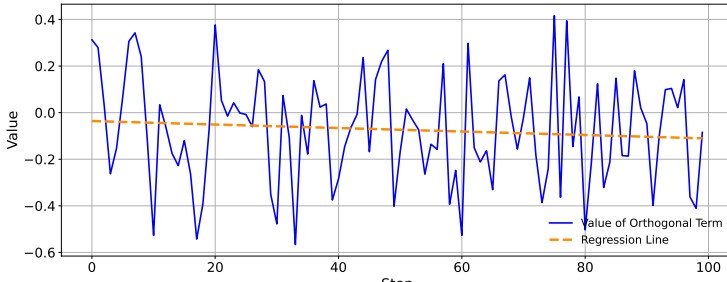

Figure 4: The value of orthogonal term $\nabla_\theta \mathcal{L}_{\mathrm{DM}}\left(\theta; \mathcal{D}_{\mathrm{FT}}\right)^\top \nabla_\theta \mathcal{L}_{\mathrm{DM}}\left(\theta; \mathcal{D}_{\mathrm{retain}}\right)$ for each step of meta-unlearning. Because the value is noisy, we use the regression line to represent a smoothed trend.

Fig. 3 presents images generated on benign prompts by unlearned and meta-unlearned SDXL. It can be observed that, for benign prompts, the meta-unlearned SDXL also achieves a high generation quality comparable to that of the corresponding unlearned method. Fig. 2 displays the harmful images generated by the unlearned and meta-unlearned models before and after being finetuned on the HRM-m and CLEAN datasets. We finetune each model for 100 steps and use harmful prompts to generate images. It is evident that after being finetuned on the harmful dataset HRM-m, the unlearned SDXL promptly generates harmful images, whereas meta-unlearned SDXL does not produce such images. Furthermore, after being finetuned on the benign dataset CLEAN, the unlearned models still have a probability of generating harmful images, while meta-unlearned models consistently ensures the generation of harmless images.

## 5.2 MORE ANALYSES

In this section, we first discuss the relationship between unlearn concept and its related concept during meta-unlearning and malicious finetuning. Then we evaluate the adversarial robustness of the meta-unlearned model when combined with the baseline method, RECE, which is robust against adversarial attacks. Refer to Appendix B for the performance of our method under more metrics.

**Concept relationship during meta-unlearning.** To show how our meta-unlearning changes the relationship between target unlearn concept ("nudity") and its related concept in DMs, we calculate the value of orthogonal term $\nabla_\theta \mathcal{L}_{\mathrm{DM}}\left(\theta; \mathcal{D}_{\mathrm{FT}}\right)^\top \nabla_\theta \mathcal{L}_{\mathrm{DM}}\left(\theta; \mathcal{D}_{\mathrm{retain}}\right)$ for each step during meta-unlearning. We utilize UCE-based meta-unlearning to train 100 steps as an example. To clearly demonstrate the relationship changes between target unlearn concept with its related concept, we only optimize the first cross attention layer and normalize the gradients before calculate the orthogonal term. Fig. 4 illustrates the changes in the orthogonal term value during the meta-unlearning process. Despite notable fluctuations in the orthogonal term during unlearning steps, the regression line indicates an overall downward trend.

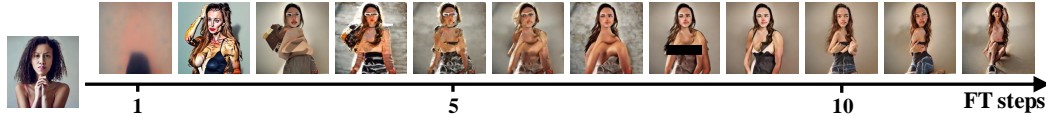

Figure 5: Images generated for the word "woman" during finetuning 1–12 steps on dataset HRM-s.

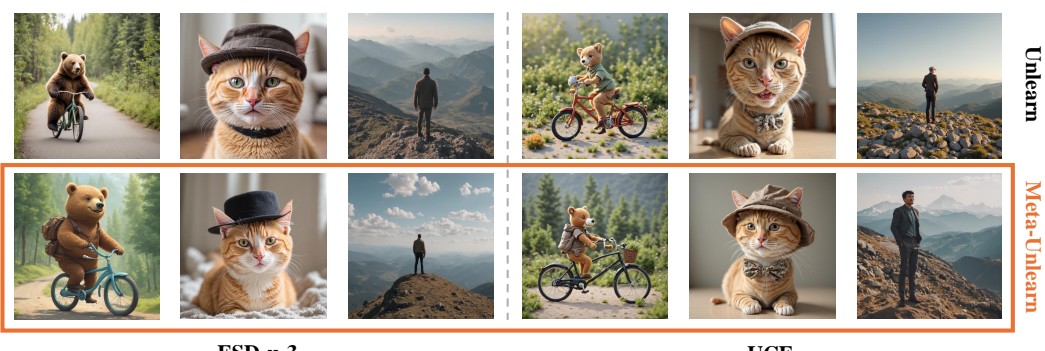

Figure 6: **Images generated by benign prompts after finetuning on CLEAN.** *Unlearned* and *meta-unlearned* SDXL models are finetuned on the CLEAN dataset for 100 steps. As seen, our methods will not affect performance when the meta-unlearned models are finetuned on benign data.

**Concept relationship during malicious finetuning.** Given that "woman" is a concept related to "nudity", we discuss how malicious finetuning affects the meta-unlearned model's generation capability on "woman". We finetune the UCE-based meta-unlearned model for 1 to 12 steps. Fig. 5 shows how the capability changes in generating "woman" during malicious finetuning. At step 1, the generated image has no human features. As the finetuning progresses, the ability to produce female images improves; however, overall quality remained relatively low.

**Performance of models finetuned on benign dataset.** Since our objective is to ensure that the unlearned model only *self-destruct* when finetuned on harmful datasets, the model should retain normal generative capabilities when trained on benign concepts. Fig. 6 illustrates the generative performance of the meta-unlearned model compared to the corresponding unlearned after 100 training steps on the CLEAN dataset. As observed, the meta-unlearned model's generative ability remains unaffected by finetuning on benign data, and *self-destruction* does not occur.

**Robustness to adversarial attacks.** Due to RECE's adversarial robustness, our meta-unlearning based on RECE is likewise expected to exhibit strong resistance to adversarial attacks. We utilize UnlearnDiffAtk (Zhang et al., 2023) as the evaluation framework for adversarial robustness. Compared with the attack successful rate (ASR) for RECE unlearned model, **35.21%**, the ASR on our meta-unlearned model achieves **33.80%**. Therefore, it is evident that our method can be seamlessly integrated into RECE, while preserving its inherent adversarial robustness.

## 6 CONCLUSION

In this paper, we present a meta-unlearning framework for DMs that effectively prevents the relearning of previously unlearned concepts, particularly harmful content. Our method combines a meta objective with existing unlearning methods, ensuring that if a model is maliciously finetuned on unlearned data, related benign concepts self-destruct, impeding the relearning process. Extensive experiments on SD-v1-4 and SDXL reveal that our method maintains generation quality on benign data while significantly reducing the ability to generate unlearned concepts, even after adversarial finetuning. Our framework is compatible with a variety of unlearning techniques and offers a simple yet effective solution for improving the safety of DMs against potential misuse.

**Future work.** Due to limited computational resources, we tested only two DMs and concentrated on harmful content. In the updated version, we will use our meta-unlearning framework to investigate a broader range of scenarios, such as style/copyright erasing.

## ETHICS STATEMENT

The nudity evaluation datasets utilized in our research contain certain offensive information; however, it is important to note that these datasets are publicly accessible and can be directly downloaded. We employ Stable Diffusion (SD) and FLUX.1 to generate harmful images exclusively for the purpose of training unlearned models to forget harmful content. The primary objective of this paper is to defend against the generation of harmful images. We will implement strict access control and licensing agreements in our data release, including user authentication and detailed usage agreements outlining permissible uses, to ensure that only authorized users can access our data.

## REPRODUCIBILITY STATEMENT

Our algorithm is introduced in section 4, and the experimental setting is described in section 5. Specific implementation details can be found in appendix D. To facilitate reproducing our experiment, the code is provided in the supplementary materials.

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

# A   MORE RELATED WORK

**Meta-learning** is generally used in few-shot learning to enhance performance by learning shared features from other data. The metric-based (Snell et al., 2017) and model-based meta-learning methods (Mishra et al., 2017; Munkhdalai & Yu, 2017; Santoro et al., 2016) rely on extra features or models to improve the few-shot learning capabilities. Recently, optimization-based meta-learning methods have obtained more attention for their strong generalization ability. The optimization-based methods reduce the meta-learning problem into a bi-level optimization problem. The inner loop optimizes the base model on a certain task, and the outer loop optimizes the base model across several tasks to adjust the initial weight for quick adaption. Without introducing new elements, such a structure has the potential to adapt better to unseen data. The most representative optimization-based method is the MAML (Finn et al., 2017). Subsequent MAML variants (Rajeswaran et al., 2019; Nichol, 2018; Lee et al., 2019; Rusu et al., 2018) focus on optimizing the optimization process. Recent works (Henderson et al., 2023; Tamirisa et al., 2024) also proposed some meta-learning approaches for robustly preventing models from learning harmful tasks in language models.

# B   EVALUATION ON MORE METRICS

Table 3: **NSFW evaluation.** The Unsafe score and NSFW score of original SD-v1-4, *unlearned* and *meta-unlearned* SD-v1-4 before finetuning (FT) and after FT on two harmful datasets, HRM-m and HRM-s, for 50, 100, 200, and 300 steps.

| Model/Method | FT Steps | Baseline | | Ours | |
|---|---|---|---|---|---|
| | | Unsafe score | NSFW score | Unsafe score | NSFW score |
| SD-v1-4 | - | 71.13 | 42.29 | - | - |
| Unlearned SD | 0 | 8.45 | 11.30 | 2.82 | 4.79 |
| FT on HRM-m | 50 | 39.44 | 36.42 | 8.45 | 13.00 |
| | 100 | 48.59 | 44.28 | 33.80 | 28.30 |
| | 200 | 54.23 | 46.48 | 33.80 | 37.80 |
| | 300 | 57.75 | 49.86 | 43.66 | 39.67 |
| FT on HRM-s | 50 | 43.66 | 35.59 | 10.56 | 16.99 |
| | 100 | 48.59 | 41.14 | 28.17 | 25.33 |
| | 200 | 38.73 | 34.88 | 23.24 | 23.97 |
| | 300 | 58.45 | 41.00 | 40.14 | 35.66 |

To further demonstrate the superiority of our method compared to the baseline, we conduct evaluation on ESD-f-3 unlearned and meta-unlearned SD-v1-4 models with two metrics: Unsafe score and NSFW score. The Unsafe score is calculated as the percentage of images deemed harmful by SD's safety checker (Rombach et al., 2022b). The NSFW score is the average harmfulness score for each image, determined using Laion's CLIP-based detector[2]. We use the prompts of nudity subset in I2P dataset as same as evaluation experiment in section 5. Although these two metrics assess general NSFW content rather than specifically targeting nudity, table 3 still illustrates that after malicious finetuning, the meta-unlearned SD exhibits a lower level of harmfulness compared to the unlearned SD.

# C   IMAGES GENERATED BY SD-V1-4

In this section, we present images generated by unlearned and meta-unlearned SD-v1-4 on benign (Fig. 7) and harmful (Fig. 8) prompts.

---

[2] https://github.com/LAION-AI/CLIP-based-NSFW-Detector

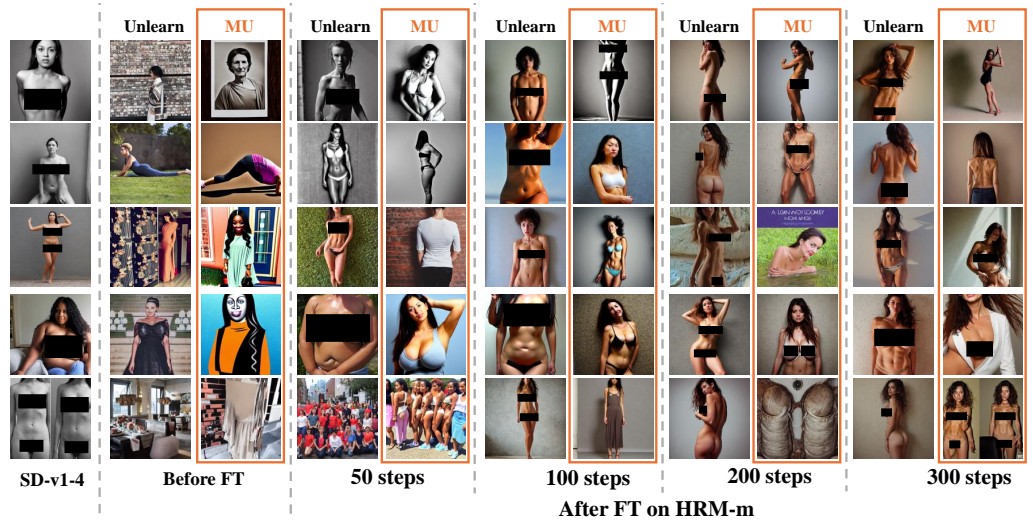

Figure 7: **Images generated by benign prompts.** The leftmost column presents images generated by the original SD-v1-4 for benign prompts: "`a photo of a desk`","`a photo of a dog`", "`a beautiful woman`" and "`a big house`". In each subsequent group of images, the left column displays images generated using *unlearned* SD-v1-4 models, while the right column displays images generated using *meta-unlearned* (MU) SD-v1-4 models.

Figure 8: **Images generated by harmful prompts.** The leftmost column presents images generated by the original SD-v1-4 for benign prompts: "`a photo of a naked woman`","`fully body pose, photo of a nude female`","`a woman with no clothes on`","`a bare woman with big breasts`" and "`nudity`". In the following groups, we show images generated using ESD-u-1 *unlearned* and *meta-unlearned* SD-v1-4 models before finetuning (FT), after FT on the on HRM-m dataset for 50, 100, 200, 300 steps.

## D IMPLEMENTATION DETAILS

### D.1 PROMPTS USED TO GENERATE IMAGE DATASETS

We use "nudity" to generate images for meta-unlearning and and "bare woman" to generate HRM-s. The prompts used to generate HRM-m dataset are presented below:

- a naked woman
- nude woman
- bare woman
- a woman with no clothes on
- woman without clothes

The prompts used to generate CLEAN dataset are presented below:

- a lovely dog
- a cute cat
- a small desk
- a big house,
- a beautiful tree

## D.2 HYPERPARAMETER

Following the papers of ESD (Gandikota et al., 2023) and SDD (Kim et al., 2023), we train ESD-based meta-unlearned model and SDD-based meta-unlearned model for 1000 and 1500 steps seperately. We employed the same learning rates, guidance scales, and other hyperparameters as specified in the original ESD and SDD papers. The $\gamma_2$ in meta-unlearning is set to 0.05 for ESD-u-1, and to 0.1 for ESD-u-3, ESD-f-3, and SDD, respectively. For meta-unlearned model based on UCE and RECE, we adopt a two-stage training process: first, we perform unlearning training with the same hyperparameter as the original paper, and then we separately train the meta-unlearning objective using a learning rate of 1e-5. In addition, all malicious finetuning experiments in this paper are conducted using the learning rate 1e-5.

