# OpenReview forum: "Meta-Unlearning on Diffusion Models: Preventing Relearning Unlearned Concepts"
_ICLR.cc/2025/Conference — ICLR 2025 Conference Withdrawn Submission_

### Official Review · Reviewer_5qaQ · 2024-11-01

**Soundness:** 4
**Presentation:** 3
**Contribution:** 3
**Rating:** 6
**Confidence:** 3

**Summary:**

The paper proposed a meta-unlearning framework to prevent the diffusion models from relearning the unlearned concepts.  Unlearning diffusion models have been studied and promising solutions have been proposed. Yet, unlearned diffusion models can relearn the unlearned concepts through fine-turning. A key innovation of this paper is that the forward perspective of how the diffusion models will be fine-tuned is considered. More specifically, the authors 'simulated' the fine-tuning process of the diffusion models and then used the 'simulated' fine-tuning process to guide the unlearning process such that (i) the learning stagnates along the direction of the unlearned concepts, and (ii) self-destruct the retained information in the diffusion models when fine-tuning. These forward perspective is novel and well motivated.

**Strengths:**

The paper provides a forward perspective of unlearning diffusion models. This approach is well motivated and sounds novel.

**Weaknesses:**

**Major arguments**:
- *A1*: The value of the orthogonal term fluctuates substantially. This can be many reasons, e.g., high dimensionality of the parameter space, lagged loss landscape, and competing objectives of L(D_{retain}) and L(D_FT). It indicates that the "self-destruct" mechanism isn't reliably working consistently. The improvement might be coming from the minimization of the gradient norm, not the "self-destruction" mechanism that the author claim. An ablation study isolating the impact of the orthogonal term would identify the key factors for the effectiveness of the proposed framework.
- *A2*: The proposed method assumes that how the diffusion model will be fine-tuned is known. In practice, however, this is rarely the case. This is a key limitation of the proposed method. The paper could be improved by showing the results for the diffusion models unlearned based on an expected fine-turning process but actually fine-tuned on a different process.
- *A3*: While the authors claim that the existing unlearning methods can relearn the unlearned concepts in their experiments, they did not show the results for all the models they claimed for (For example, Wu, Yongliang, et al. "Unlearning Concepts in Diffusion Model via Concept Domain Correction and Concept Preserving Gradient." arXiv preprint arXiv:2405.15304 (2024)). It is not clear whether the claims are valid without the results.

**Some technical comments**:
- In lines 6 and 10 in the Algorithm 1, I think we should substract the gradient instead of adding it in order to minimize the objective.

**Questions:**

1. To what extent the orthogonal term improves the unlearning performance?
2. When the diffusion models are fine-tuned differently from what anticipated at unlearning process, how effective is the proposed method?
3. Provide evidence for the claim as described in A3.

---

### Official Review · Reviewer_TGTA · 2024-11-02

**Soundness:** 2
**Presentation:** 2
**Contribution:** 2
**Rating:** 3
**Confidence:** 4

**Summary:**

This paper discuss the phenomenon of "malicious fine-tuning" in diffusion models and introduces a meta-unlearning method to mitigate it. The proposed approach prevents the model from relearning malicious concepts by "self-destructing" related benign concepts that could support the undesired relearning.

**Strengths:**

1. Introducing meta-learning into the unlearning domain is an interesting approach.
2. The extensive experimental results demonstrate the effectiveness of the proposed method.

**Weaknesses:**

1. The problem formulation is unclear to me. What exactly is meant by "malicious fine-tuning"? Does this refer to fine-tuning on a malicious dataset D_{FT} \in D_{forget}? Is "fine-tuning" here simply standard fine-tuning, or fine-tune based unlearning (e.g., gradient ascent)? If malicious fine-tuning simply involves standard fine-tuning on a subset of D_{forget}, then it’s somewhat expected that the model would relearn knowledge intended for unlearning, which makes the problem formulation seem a bit trivial.

2. I’m also skeptical of the approach that destructs related benign concepts. In my opinion, the goal should be to unlearn harmful concepts (e.g., "nudity") while retaining benign knowledge (e.g., "skin"), rather than destruct it. Effective unlearning of a harmful concept should not come at the cost of unlearning related benign concepts, unless absolutely necessary—which I think it is not.

3. For the "Robustness to adversarial attacks" analysis, while it's reasonable to compare with RECE given that your approach builds on it, it would strengthen the work to include comparisons with other baselines.

**Questions:**

1. In ‘Concept relationship during meta-unlearning’ analysis, the orthogonal term appears to show a slight downward trend. If I interpret this correctly, it suggests that the target unlearned concept (e.g., "nudity") and the related benign concept are becoming decoupled. Is this result expected, or is it a side effect of the meta-unlearning approach? Since the goal is to unlearn the malicious concept and destruct the knowledge of the related concept, why are they being decoupled?

2. Could you provide more detail for the ‘robustness to adversarial attacks’ analysis? Specifically, could you describe the type of adversarial attack used and share implementation details? I didn’t find this information in either the main paper or the appendix, but please let me know if I’ve overlooked something.

---

### Official Review · Reviewer_SSbC · 2024-11-04

**Soundness:** 3
**Presentation:** 1
**Contribution:** 2
**Rating:** 3
**Confidence:** 3

**Summary:**

This paper describes a novel approach to unlearning of concepts in (image) diffusion models that aims towards robustness against re-learning those same concepts.

**Strengths:**

This paper addresses a relatively novel, interesting problem. The proposed solution appears to work well in the evaluated scenarios (-> see weaknesses) and is applicable to many models, concepts and unlearning approaches.

Given my background, I cannot judge the completeness of related work. However, it is described in detail and cites also recent advances.

**Weaknesses:**

My main concern with the paper is that during the unlearning the knowledge of the finetuning dataset $D_{FT}$ is assumed. For a practical scenario, I would rather expect that the re-learning could take place with any set of images with the respective concepts. As a consequence such images cannot be directly "meta-unlearned" against. It would be crucial to see how the relearning performs on such images.

The evaluation and results is not described well: On which images have the scores of Table 1 been computed? For the table, please add also standard deviation. In this scenario it seems plausible (though not probable) that there might be outliers appearing for meta-unlearning. How is the similarity of the generated images to the finetune dataset? Also, the evaluation is also done on only one setting. (nudity). Additionally, how close are the generated images to the finetuning datasets? From my understanding, it would be impossible to avoid having 1-1 copies of the fine-tuning dataset in the image generation.

The presentation of the paper should be improved considerably. Some (partially subjective) suggestions:
- Figure 1. did confuse me more that it helped. I would also discourage the forward reference here.
- The description of related work contains a lot of citations but they are not explained well. I would focus on a smaller area of interest but invest more space into covering this more in-depth. In particular, consider removing Section 2.2 and expland Section 2.3,

**Questions:**

How does the approach perform if the finetuning set is unknown?

---

### Author Response · Authors · 2024-11-15

Dear Reviewers,

We appreciate your time and insightful feedback on our work.

A publicly available unlearned DM could be maliciously finetuned to relearn the unlearned concepts. In our meta-unlearning framework, we have two design goals: (1) a meta-unlearned DM should behave like an unlearned DM when used as is; and (2) if someone maliciously finetunes the meta-unlearned DM, the model should **automatically self-destruct to avoid misuse**.

Our meta-unlearning framework is compatible with almost all existing unlearning methods, requiring only the addition of an easy-to-implement meta objective. When combining adversarially robust unlearning methods, we can defend against both adversarial attacks and malicious finetuning.

After careful consideration, we have decided to withdraw our paper from ICLR. Thank you once again for your thorough review and thoughtful comments.

Best,\
The Authors

---

### Note · Authors · 2024-11-15

I have read and agree with the venue's withdrawal policy on behalf of myself and my co-authors.